# Predicting patients with false negative SARS-CoV-2 testing at hospital admission: A retrospective multi-center study

**Lama Ghazi**[ID][☯]**, Michael Simonov**[ID][☯]**, Sherry G. Mansour, Dennis G. Moledina**[ID]**, Jason H. Greenberg, Yu Yamamoto, Aditya Biswas, F. Perry Wilson**[ID]*

Clinical and Translational Research Accelerator, Yale University, New Haven, Connecticut, United States of America

☯ These authors contributed equally to this work.
* francis.p.wilson@yale.edu

**Data Availability Statement:** The data used for this manuscript contains time-updated de-identified information for several years of hospitalized patient

## Abstract

### Importance

False negative SARS-CoV-2 tests can lead to spread of infection in the inpatient setting to other patients and healthcare workers. However, the population of patients with COVID who are admitted with false negative testing is unstudied.

### Objective

To characterize and develop a model to predict true SARS-CoV-2 infection among patients who initially test negative for COVID by PCR.

### Design

Retrospective cohort study.

### Setting

Five hospitals within the Yale New Haven Health System between 3/10/2020 and 9/1/2020.

### Participants

Adult patients who received diagnostic testing for SARS-CoV-2 virus within the first 96 hours of hospitalization.

### Exposure

We developed a logistic regression model from readily available electronic health record data to predict SARS-CoV-2 positivity in patients who were positive for COVID and those who were negative and never retested.

data. These data, though de-identified with respect to unique patient identifiers (e.g. medical record numbers) contain sensitive data such as admission/discharge dates, times of medication administrations, and demographic information. These data are sensitive and privileged thus we are not able to freely share this data. The authors are committed to openly and freely sharing data with researchers who are willing to aabide by a data use agreement. We would like to provide the contact information for the Yale Human Research Protection Program (HRPP) through which data use agreements may be forwarded. The office may be reached via email through HRPP@yale.edu, via phone at 203-785-4688, and fax at 203-785-2847. The mailing address of this office is PO Box 208327, New Haven, CT 06520-8327.

**Funding:** FPW has funding through grants R01DK113191 and P30DK079310 from the National Institute of Diabetes and Digestive and Kidney Diseases. Other authors received no specific funding for this work.

**Competing interests:** The authors have declared that no competing interests exist.

## Main outcomes and measures

This model was applied to patients testing negative for SARS-CoV-2 who were retested within the first 96 hours of hospitalization. We evaluated the ability of the model to discriminate between patients who would subsequently retest negative and those who would subsequently retest positive.

## Results

We included 31,459 hospitalized adult patients; 2,666 of these patients tested positive for COVID and 3,511 initially tested negative for COVID and were retested. Of the patients who were retested, 61 (1.7%) had a subsequent positive COVID test. The model showed that higher age, vital sign abnormalities, and lower white blood cell count served as strong predictors for COVID positivity in these patients. The model had moderate performance to predict which patients would retest positive with a test set area under the receiver-operator characteristic (ROC) of 0.76 (95% CI 0.70–0.83). Using a cutpoint for our risk prediction model at the 90th percentile for probability, we were able to capture 35/61 (57%) of the patients who would retest positive. This cutpoint amounts to a number-needed-to-retest range between 15 and 77 patients.

## Conclusion and relevance

We show that a pragmatic model can predict which patients should be retested for COVID. Further research is required to determine if this risk model can be applied prospectively in hospitalized patients to prevent the spread of SARS-CoV-2 infections.

## Introduction

Coronavirus disease-2019 (COVID-19), the illness caused by the SARS-CoV2 virus has had widespread global effects and has caused significant strain on both inpatient and outpatient healthcare institutions [1, 2]. Reports during the early phase of the pandemic showed significant nosocomial transmission of disease [3–5]. Therefore, a major consideration for health systems is mitigating the spread of virus within the hospital setting to uninfected patients and to healthcare workers. Another unique challenge of COVID-19 has been management of protective personal equipment and maintaining adequate rooming and facilities for patients hospitalized with the illness [6].

Many hospitals have enacted strategies to test patients directly in the emergency room prior to admission to a hospital unit with the goal of appropriately rooming COVID-positive patients on COVID-specific wards and provide appropriate personal protective equipment to healthcare workers [7]. One unstudied yet important population are patients who initially test negative for COVID and later retest positive for the virus [8]. Though COVID tests used in hospital settings are very specific, sensitivity is much lower with significant temporal variability of viral shedding; moreover, a recent systematic review reports a false negative rate of 13%, a number sufficiently high to be clinically meaningful [9–11]. Such patients may pose a significant risk especially in the hospital setting. These patients may be roomed with non-infected patients and thus may expose other patients, visitors, and healthcare workers to SARS-CoV-2. Moreover, nosocomial SARS-CoV-2 infections in hospitalized patients are concerning as

hospitalized patients are often older, immunocompromised, and have multiple comorbidities which are all risk factors for severe COVID [12].

In this retrospective study, we evaluate this group of patients who initially test COVID negative per nasopharyngeal polymerase chain reaction (PCR) testing but subsequently retest positive to identify patient characteristics, vital signs, and laboratory tests that may predict a subsequent positive test for COVID. We develop a risk model for predicting a patient's COVID 'positivity' and apply it to the broader COVID-negative cohort to identify patients who will later have a positive test. We hypothesized that a model could be developed that would discriminate which patients who initially test negative for COVID may indeed have the infection, identifying a population for targeted re-testing.

## Methods

### Patients and setting

We included adult patients hospitalized at one of five hospitals within the Yale New Haven Health System (YNHHS) between 3/10/2020 and 9/1/2020 who received nasopharyngeal PCR testing for SARS-CoV-2 virus during the time period of their hospitalization. SARS-CoV-2 tests included several multiplex real time RT-PCR tests (GeneXpert–Cepheid; Siplexa–Diasorin; TaqPath–Thermo Fisher), transcription mediated amplification test (Panther—Hologic) and a singleplex real time RT PCR test (CDC–lab developed). Data regarding specific test used for each sample were not available for this study. YNHHS includes 6 hospitals across Connecticut and Rhode Island and includes a variety of settings, including academic/community, urban/sub-urban, and teaching/non-teaching.

The first 96 hours of a patient's hospitalization served as the observation period with the aim of limiting the analyses to patients who likely initially had COVID on presentation rather than patients who developed nosocomial COVID during their hospitalization. Patients who did not have any COVID tests during the observation period were excluded from analysis.

This study operated under a waiver of informed consent and was approved by the Yale Human Investigation Committee (HIC # 2000027733).

### Variables and outcomes

We collected longitudinal data from the electronic health record including demographics, comorbidities, procedures, medications, laboratory results, and vital signs. All data were extracted from the data warehouse of our electronic health record vendor Epic (Verona, WI).

Patient variables were chosen pragmatically for those that would be simpler to embed into a clinical decision support platform either directly onto the EHR or as a web service. These variables were chosen as they contained very low (<10%) missingness for hospitalized patients within the first 24 hours of hospitalization. Variables included in the model included demographics (age, sex, race), comorbidities (congestive heart failure, chronic pulmonary disease, diabetes, obesity, history of arrhythmia, hypertension, alcohol use disorder, metastatic cancer, stroke, transient ischemic attack, HIV, and the Elixhauser comorbidity index), laboratory values (sodium, potassium, chloride, bicarbonate, blood urea nitrogen, creatinine, glucose, hemoglobin, platelet count, white blood cell count and lymphocyte percentage) and vital signs (temperature, systolic blood pressure, diastolic blood pressure, respiratory rate, and oxygen saturation). Comorbidities were defined as per the Elixhauser comorbidity index based on codes from the International Classification of Diseases-10 [13]. The first measurement for these variables were used in analyses.

## Statistical methods

We used descriptive statistics to compare the populations of patients who initially tested positive, those who initially tested negative and later tested positive, and those who initially tested negative and remained negative throughout the hospitalization. Chi-square testing was used to compare categorical variables and the Kruskall-Wallis test was used for continuous covariates.

We trained a logistic regression model to predict COVID-positivity in patients with an initial positive COVID test (+/0) and those with an initial negative COVID test who were never retested (-/0). We then tested the performance of this model amongst individuals with an initial negative COVID test who were retested and negative (-/-) and retested and positive (-/+) within the first 96 hours of their hospitalization. This allowed evaluation of model performance among individuals that could clearly be classified as 'false negative' or 'true negative' at the time of initial testing. Variable importance in the logistic regression model were determined by the magnitude of the absolute value of the z-score.

Area under the operator receiver curve (AUROC) as well as the precision-recall curve (PRC) are reported regarding performance of the model on the validation set. Quantiles of probabilities from the logistic model were developed from the training set and then applied to test set probabilities to determine cut points for the prediction. We report quantile of probability which was chosen clinically to optimize the sensitivity of patients who would be appropriately identified as indeed having COVID while minimizing the 'number needed to test'.

All analyses were performed using R (Version 4.0.0, Vienna, Austria) [14]. Logistic regression models were developed using the *glm* function from the 'stats' package in R. We defined statistical significance at $P < 0.05$.

This study utilized the Strengthening the Reporting of Observation Studies in Epidemiology (STROBE) guidelines.

## Results

There were a total of 40,030 patients hospitalized at the five Yale-New Haven Health system hospitals between 3/10/2020 and 9/1/2020. Of these, 31,459 adult patients had a COVID test during the first 96 hours of hospitalization and were included in analyses (**Fig 1**). Of these patients, there were 2,666 patients who tested positive for COVID and 25,382 patients who tested negative and were never retested. This group of 28,048 patients served as the training population for modeling. The validation set was composed of 3,511 patients who initially tested negative for COVID and were retested, of which 61 (1.7%) retested positive.

We compared patients who were initially COVID-positive to those who were falsely negative on for their initial test (**Table 1**). These two populations were similar in terms of demographics, baseline vital signs, comorbidities, as well as initial laboratory values. On admission, COVID-negative patients were noted to have a higher Elixhauser comorbidity score, more diabetes, slightly elevated creatinine, and slightly lower hemoglobin. Characteristic of all patients are presented in **S1 Table**. Manual chart inspection was performed for the 61 patients who retested positive; reasons for subsequent test included high clinical suspicion despite negative test (51%), testing as part of disposition planning (5%) or prior to undergoing a procedure (7%), testing prior to hospital transfer (3%), inconclusive first COVID test (2%), as well as unclear reason for testing (31%). Clinical suspicion included a wide variety of symptoms and findings including abnormal imaging, new-onset fever, hypoxia, shortness of breath, and known contact with a patient with COVID-19. 40% of patients who retested positive did not have symptoms on admission. The mean number of days between first and second test was 2.5 days (IQR: 1–2 days).

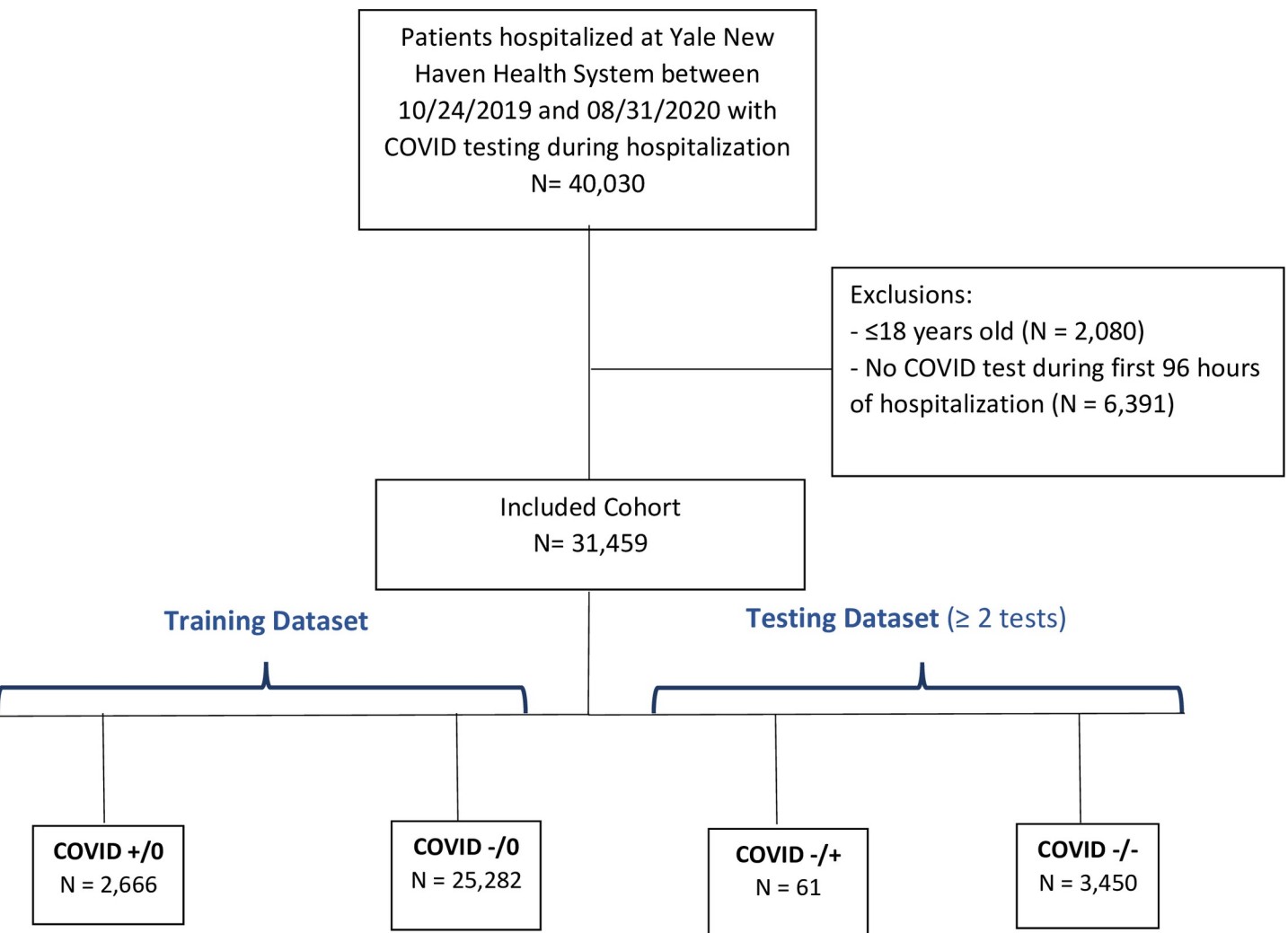

**Fig 1. Cohort diagram.** COVID +/0: tested positive for COVID on admission and was not retested within 4 days. COVID -/0: tested negative for COVID on admission and was not retested within 4 days. COVID -/+: tested positive for COVID on admission and upon retesting (within 4 days) tested positive for COVID. COVID -/-: tested negative for COVID on admission and upon retesting (within 4 days) tested negative for COVID.

A multivariable logistic regression to predict initial COVID positivity was performed with the full equation of the model with covariates supplied in in **S1A and S1B Fig**. The most important variables in the logistic regression, as measured by the absolute value of their z-score, to predict increased risk of COVID positivity were higher age, black race, lower initial oxygen saturation, higher initial temperature, and lower white blood cell count.

The model was then applied to predict which patients would retest as COVID positive in the validation cohort. The AUROC of the model to predict this outcome was 0.76 (95% CI 0.70–0.83) with AUROC curve displayed in **Fig 2**. The precision-recall curve is provided in **S2 Fig**.

The probability scores from the logistic regression model for the several patient groups of patients who were initially COVID negative and not retested (-/0), COVID negative and retested negative (-/-), COVID negative and retested positive (-/+), and COVID positive and not retested (+/0) are displayed in **Fig 3**. Patients categorized as false negatives on initial testing had higher probabilities per the model than the persistently COVID negative cohort.

**Table 1. Characteristics of patients with COVID positive test on admission vs. those who tested negative on admission and had a subsequent COVID positive test.**

| | COVID (+/0) | COVID (-/+) | |
|---|---|---|---|
| | N = 2,666 | N = 61 | p-value[1] |
| **Demographics** | | | |
| Age, median (IQR), years | 66.01 (52.03, 79.99) | 65.85 (53.85, 77.34) | >0.9 |
| Female, No/Total No (%) | 1,332 / 2,666 (50) | 26 / 61 (43) | 0.3 |
| Race, No/Total No (%) | | | 0.5 |
| Non-White | 1,407 / 2,666 (53%) | 29 / 61 (48%) | |
| White | 1,259 / 2,666 (47%) | 32 / 61 (52%) | |
| Latino, No/Total No (%) | 691 / 2,666 (26%) | 14 / 61 (23%) | 0.7 |
| Hospital, No/Total No (%) | | | 0.4 |
| Yale New Haven Hospital | 871 / 2,666 (33%) | 22 / 61 (36%) | |
| St Raphael's Campus | 542 / 2,666 (20%) | 13 / 61 (21%) | |
| Bridgeport Hospital | 739 / 2,666 (28%) | 19 / 61 (31%) | |
| Greenwich Hospital | 421 / 2,666 (16%) | 6 / 61 (9.8%) | |
| Lawrence and Memorial Hospital | 76 / 2,666 (2.9%) | 0 / 61 (0%) | |
| Westerly Hospital | 17 / 2,666 (0.6%) | 1 / 61 (1.6%) | |
| **Baseline Characteristics, median (IQR)** | | | |
| Systolic, mmHg | 131.00 (117.00, 148.00) | 125.00 (108.00, 147.00) | 0.07 |
| Diastolic, mmHg | 76.00 (65.00, 85.00) | 75.00 (66.00, 82.00) | 0.4 |
| Pulse, beats per minute | 93.00 (80.00, 108.00) | 90.00 (82.00, 112.00) | 0.8 |
| Respiratory rate, breaths per minute | 20.00 (18.00, 22.00) | 20.00 (18.00, 22.00) | >0.9 |
| spO2, % | 96.00 (93.00, 98.00) | 96.00 (94.00, 98.00) | 0.3 |
| Temperature, Fahrenheit | 98.74 (97.89, 100.38) | 98.74 (97.84, 100.26) | >0.9 |
| BMI, Kg/m$^2$ | 28.37 (24.15, 33.82) | 27.39 (24.06, 30.57) | 0.2 |
| **Comorbidities** | | | |
| Elixhauser score, median (IQR) | 5.00 (2.00, 9.00) | 6.00 (3.00, 11.00) | 0.04 |
| CHF, No/Total No(%) | 644 / 2,666 (24%) | 17 / 61 (28%) | 0.6 |
| CPD, No/Total No(%) | 882 / 2,666 (33%) | 21 / 61 (34%) | >0.9 |
| Diabetes, No/Total No(%) | 1,069 / 2,666 (40%) | 35 / 61 (57%) | 0.001 |
| Obesity, No/Total No(%) | 853 / 2,666 (32%) | 23 / 61 (38%) | 0.4 |
| Arrhythmia, No/Total No(%) | 1,006 / 2,666 (38%) | 26 / 61 (43%) | 0.5 |
| HTN, No/Total No(%) | 1,744 / 2,666 (65%) | 42 / 61 (69%) | 0.7 |
| Malignancy, No/Total No(%) | 305 / 2,666 (11%) | 7 / 61 (11%) | >0.9 |
| Metastasis, No/Total No(%) | 77 / 2,666 (2.9%) | 3 / 61 (4.9%) | 0.4 |
| Alcohol abuse, No/Total No(%) | 271 / 2,666 (10%) | 7 / 61 (11%) | >0.9 |
| Drug abuse, No/Total No(%) | 258 / 2,666 (9.7%) | 6 / 61 (9.8%) | >0.9 |
| Stroke, No/Total No(%) | 155 / 2,666 (5.8%) | 7 / 61 (11%) | 0.09 |
| TIA, No/Total No(%) | 55 / 2,666 (2.1%) | 2 / 61 (3.3%) | 0.4 |
| HIV, No/Total No(%) | 40 / 2,666 (1.5%) | 1 / 61 (1.6%) | 0.6 |
| **Laboratory Values, median (IQR)** | | | |
| Sodium, mmol/L | 137.00 (134.00, 140.00) | 138.00 (136.00, 143.00) | 0.8 |
| Potassium, mmol/L | 4.00 (3.70, 4.40) | 4.10 (3.70, 4.30) | 0.4 |
| Bicarbonate, mmol/L | 24.40 (22.00, 27.00) | 24.00 (22.00, 26.25) | 0.2 |
| BUN, mg/dL | 18.00 (12.00, 30.00) | 20.00 (13.00, 34.00) | 0.3 |
| Creatinine, mg/dL | 1.00 (0.78, 1.47) | 1.08 (0.81, 1.60) | 0.014 |
| Chloride, mmol/L | 100.00 (97.00, 104.00) | 102.00 (98.00, 107.00) | 0.4 |
| Glucose, mmol/L | 122.00 (104.00, 162.00) | 125.00 (109.00, 161.00) | 0.2 |

(*Continued*)

**Table 1.** (Continued)

| | COVID (+/0) | COVID (-/+) | |
|---|---|---|---|
| | N = 2,666 | N = 61 | p-value[1] |
| Hemoglobin, g/dL | 12.90 (11.50, 14.30) | 12.50 (10.50, 14.00) | 0.004 |
| Platelet count, x10⁹/L | 204.00 (159.00, 263.00) | 234.00 (190.00, 327.00) | 0.003 |
| WBCC, x10³ mm³ | 6.80 (5.10, 9.40) | 8.60 (5.80, 11.70) | 0.3 |
| % Lymphocyte, % | 15.10 (9.40, 22.10) | 13.90 (10.20, 17.80) | >0.9 |
| Anion Gap. mmol/L | 14.00 (12.00, 16.00) | 14.00 (12.00, 16.00) | 0.3 |

[1]Statistical tests performed: Kruskal-Wallis test; chi-square test of independence; Fisher's Exact Test for Count Data with simulated p-value (based on 2000 replicates); Fisher's exact test

Systolic: systolic blood pressure; Diastolic: diastolic blood pressure; Pulse: pulse rate (beats per minute); Respiratory rate; spO2: oxygen saturation; Temperature; BMI: body mass index, CHF: congestive heart failure; CPD: chronic pulmonary disease; HTN: hypertension; TIA; transient ischemic attack; HIV; human immunodeficiency virus

SI conversion factors: for BUN multiply by 0.357 (mmol/L); for creatinine multiply by 88.4 (micromol/L); for WBCC x10³ mm3 is equivalent to liter

Probability of testing positive (mean, 95% CI) for COVID among the COVID (-/0), COVID (-/-), COVID (-/+) and COVID (+/0) was 0.077 (0.075, 0.078), 0.10 (0.09, 0.11), 0.28 (0.21, 0.35) and 0.34 (0.33, 0.36) respectively.

Based on the precision-recall curve, a cutpoint of >90th percentile for the probability per the logistic model was used as the predictor for whether a patient who initially tested negative

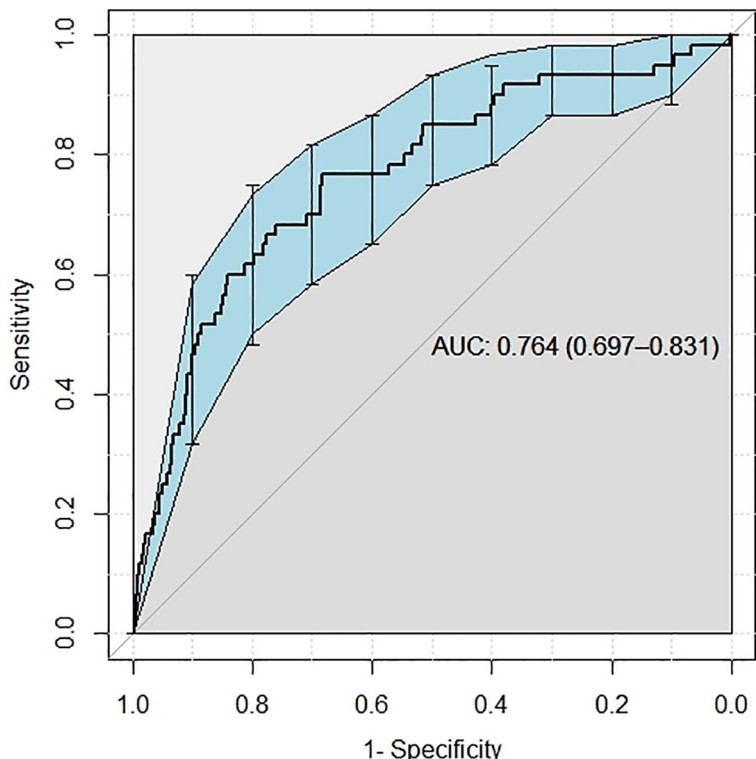

**Fig 2. Receiver operator curve to detect COVID test positivity among those who had a negative COVID test on admission and were retested.**

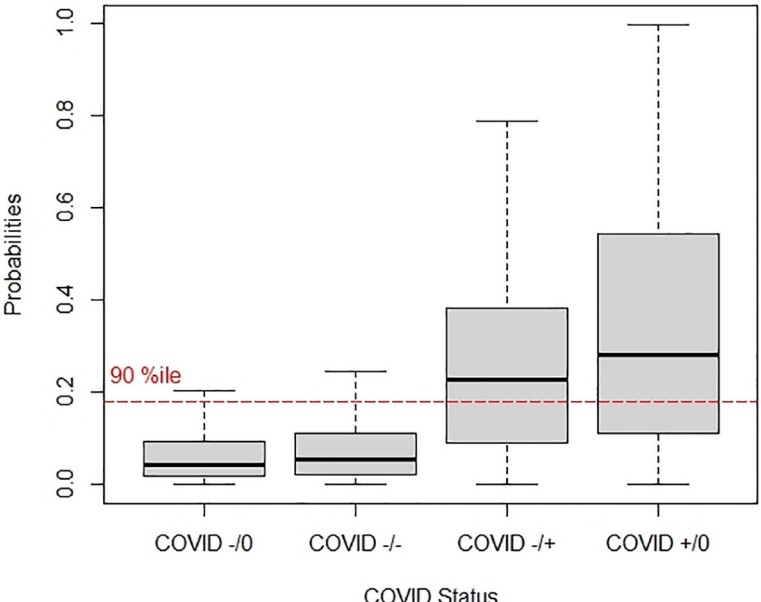

**Fig 3. Prediction model probabilities of testing positive for COVID among subpopulations.** COVID +/0: tested positive for COVID on admission and was not retested within 4 days. COVID -/0: tested negative for COVID on admission and was not retested within 4 days. COVID -/+: tested positive for COVID on admission and upon retesting (within 4 days) tested positive for COVID. COVID -/-: tested negative for COVID on admission and upon retesting (within 4 days) tested negative for COVID.

for COVID would retest positive. At this cutpoint, the model predicts that 536 patients in the validation cohort are COVID positive; 35/536 were indeed COVID positive on retest (6.5%) or one of every 15 patients; notably this would capture 57% of the total false negative patients. If this model threshold is applied over all initially COVID negative patients, 35/2,680 (1.3%) would be captured, equating to one true positive per 77 tests.

## Discussion

In this study, we assessed the performance of a model for predicting which patients who are initially deemed COVID-negative may retest positive. Our model used variables which are routinely measured for hospitalized patients and displayed good performance to discriminate which patients, when retested, would retest positive. Several variables appeared important for predicting which patients may need to be retested for COVID; increased age, lower oxygen saturation, higher temperature, and lower white blood cell count were associated with COVID positivity. These predictive variables are concordant with previous models of COVID positivity [15, 16].

We chose a cutpoint of model risk prediction that maximized the sensitivity of patients correctly identified while minimizing the number of patients who would need to be tested. At the 90[th] percentile of model risk score, we determined a 'number needed to test' ranging from best to worst case scenario of 15 to 77 patients, respectively. The worst case assumes the unlikely scenario where zero of the patients who initially tested negative and never retested (-/0) truly had COVID; thus, the true number needed to test is very likely lower than this upper bound.

Our study has several strengths. First, our model was built and tested on a very large patient dataset with data from 6 hospitals capturing a broad diversity of patients and clinical settings. Second, we used readily available data elements from the EHR which promotes ease of integration of such a model, rather than more complicated modeling approaches which may require

non-EHR solutions such as cloud computing to apply. Our model does not require measurement of biomarkers, cytokines, or other specialized clinical measurements. Third, our model had robust performance despite being trained over a very broad population of hospitalized adults with COVID tests and was validated in a fundamentally different population than that in which it was derived. We argue that the model is thus broadly generalizable for hospitalized patients. For ready deployment of the model, institutions may apply the model formula presented in **S1A Fig** and selecting a cutpoint that aligns with the goals and testing capabilities of the institution (as per above we highlight a cutpoint at the 90th percentile).

Our study should be viewed in light of several weaknesses. First, our risk model demonstrated moderate performance, thus we do acknowledge that many patients would need to be retested to find a single COVID positive patient. Second, our model was built from and applied to patients who had vital signs, a basic metabolic panel, and a complete blood count measured on admission; thus the model would not be generalizable to patients who may not have vital signs or laboratory values obtained (e.g psychiatric patients or routine obstetric patients). Third, our study is retrospective in nature and we are unable to conclude the efficacy of the implementation of this model for retesting. Another limitation is that our model was evaluated on patients who were tested twice for COVID; there were many patients who were COVID negative on presentation and never retested, therefore we are unable to provide a clear number-needed-to-test as some of these patients may have been false negatives.

We propose that by building and embedding a model using variables commonly available in the EHR, hospitals could flag patients for targeted retesting, potentially reducing nosocomial spread of COVID-19. Testing between 15 and 77 patients to find a single COVID negative patient who is truly positive should be considered in light of several logistic concerns. On one hand, this is a large amount of testing which may bring about issues of false positive COVID tests and significant expenditure of resources. Conversely, if a health system has ample COVID testing capabilities or capabilities to consider pooled COVID testing, this approach may be reasonable. We also argue that the effects of missed COVID positive patients may be profound at an institution with potential infection of other patients within a ward or infection of healthcare workers and other hospital staff who may believe the patient is 'ruled out' for COVID. Further investigation is warranted to determine the cost effectiveness of an algorithm-guided retesting approach.

## Conclusions

Our study is the first description of and model development for patients who are initially tested negative for COVID on hospitalization but are later retested and found to be COVID positive. We show that a pragmatic model can be constructed to predict which patients should be retested for COVID and found a reasonable number-needed-to-test between 15 and 77 hospitalized patients. Further research is needed to determine the cost-effectiveness of implementing a retesting approach as well as its efficacy in clinical practice.

## Supporting information

**S1 Fig. A.** Closed form equation and coefficients for regression coefficients model to predict probability of having a COVID positive test. Bolded values are significant predictors; hospital (range 1–6: 1-Yale New Haven Hospital, 2: St Raphael's Campus, 3: Bridgeport Hospital, 4: Greenwich Hospital, 5: Lawrence and Memorial Hospital, 6: Westerly Hospital); BP: blood pressure. **B.** Z-scores of model covariates.
(TIF)

**S2 Fig. Precision/Recall curve to detect COVID positivity among those who had a negative COVID test on admission and were retested.** Precision is the positive predictive value of having a COVID positive test and recall is the sensitivity of having a positive COVID test. Sensitivity in our testing cohort is 0.57 and the positive predictive value is 0.61 as can be seen on this graph.
(TIF)

**S1 Table. Demographic and clinical patient characteristics in cohort by COVID testing status.**
(DOCX)

## Author Contributions

**Conceptualization:** Lama Ghazi, Michael Simonov, Sherry G. Mansour, Dennis G. Moledina, Jason H. Greenberg, Aditya Biswas, F. Perry Wilson.

**Data curation:** Michael Simonov, Yu Yamamoto, F. Perry Wilson.

**Formal analysis:** Lama Ghazi, Michael Simonov, Yu Yamamoto, F. Perry Wilson.

**Funding acquisition:** F. Perry Wilson.

**Investigation:** Michael Simonov, F. Perry Wilson.

**Methodology:** Lama Ghazi, Michael Simonov, Sherry G. Mansour, Dennis G. Moledina, Jason H. Greenberg, Yu Yamamoto, Aditya Biswas, F. Perry Wilson.

**Project administration:** F. Perry Wilson.

**Resources:** F. Perry Wilson.

**Software:** F. Perry Wilson.

**Supervision:** F. Perry Wilson.

**Visualization:** Lama Ghazi, Michael Simonov.

**Writing – original draft:** Lama Ghazi, Michael Simonov, F. Perry Wilson.

**Writing – review & editing:** Lama Ghazi, Michael Simonov, Sherry G. Mansour, Dennis G. Moledina, Jason H. Greenberg, Yu Yamamoto, Aditya Biswas, F. Perry Wilson.

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
