## [Decision Letter · Decision Letter 0]

6 Apr 2021

PONE-D-20-38011

Predicting patients wi­­­th false negative SARS-CoV-2 testing at hospital admission: A retrospective multi-center study

PLOS ONE

Dear Dr. Simonov,

Thank you for submitting your manuscript to PLOS ONE. After careful consideration, we feel that it has merit but does not fully meet PLOS ONE’s publication criteria as it currently stands. Therefore, we invite you to submit a revised version of the manuscript that addresses the points raised during the review process.

It is an interesting paper.We suggest to better clarify the rationale of the investigation.

We look forward to receiving your revised manuscript.

Kind regards,

Chiara Lazzeri

Academic Editor

PLOS ONE

Journal Requirements:

3) In your Methods section, please provide additional information about the methodology used (the equations representing the model; how the model was calibrated; and what parameters were applied).

4)  Thank you for stating the following in the Funding/Support Section of your manuscript:

[R01DK113191 and P30DK079310 to FPW]

 [The author(s) received no specific funding for this work.]

Reviewers' comments:

Reviewer's Responses to Questions

**Comments to the Author**

1. Is the manuscript technically sound, and do the data support the conclusions?

Reviewer #1: Partly

2. Has the statistical analysis been performed appropriately and rigorously? 

Reviewer #1: Yes

3. Have the authors made all data underlying the findings in their manuscript fully available?

Reviewer #1: Yes

4. Is the manuscript presented in an intelligible fashion and written in standard English?

Reviewer #1: Yes

5. Review Comments to the Author

Reviewer #1: The study has very interesting premises aiming to find a simple system to identify false negative COVID-19 patients. However, there are some issues that in my opinion, should be addressed. As a matter of fact, intermittent positive PCR tests for SARS-CoV-2 is a common evenience but the viability of the virus in those cases is still controversial. The persistence of SARS-CoV-2 and the development of intermittent positive tests may merely indicate the detection of viral RNA. The prolonged shedding of the virus is not an indication of continued infection.

For this reasons, a more accurate evaluation of the false negative population in the study should be provided otherwise the premises for the whole subsequent analysis fail. In this perspective, authors should add some valuable information (at least for the 61 (+/-) patients) like the days from the onset of symptoms to the swab, the reason for hospitalization, SARS-CoV-2 serology, and the immunological status of the included patients (an immunocompromised patient prolonged viral shedding). Would be interesting also understand the reasons for re-testing and the allocation of those patients in the time within the re- test (covid- or no covid ward). Finally in the methods authors should include all types of RT-PCR kits, regardless and the brand or manufacturer, the extraction method used, the number of target gene assays assessed, or the cycle threshold value 136 for positivity.

Minor issues:

Authors in the introduction should clearly state which test are they talking about.

“… approximately 30% of test in COVID patients are false negative…”. It looks a very high percentage. This data is controversial. Several studies report a lower rate. Authors should explain that and mitigate this affirmation.

Please provide P value in the table with same number of decimals

In my opinion a table in the main text that clearly indicates how to identify the patients who are most likely to be positive in a re-test is missing. The table would help the reader and add clear practical value to the work.

“……We argue that the model is thus broadly generalizable”… In my opinion authors should specify that this model could be generalizable only for hospitalized patients.

“….To our knowledge, this is the first study to investigate the population of false negative patients with

COVID in the hospitalized setting..” . Indeed, there are some studies analysing false positive patients proposing algorithm to identify them (es. Laboratory markers included in the Corona Score can identify false negative results on COVID-19 RT-PCR in the emergency room. Roberto Assandri et al.). I would re-phrase maybe emphasising that you are not studying false negative, but you are proposing a new model

6. PLOS authors have the option to publish the peer review history of their article (what does this mean?). If published, this will include your full peer review and any attached files.

Reviewer #1: No

---

## [Author Response · Author response to Decision Letter 0]

22 Apr 2021

Please review the response to reviewers document attached as per this submission.

---

## [Editor Report · Decision Letter 1]

26 Apr 2021

Predicting patients wi­­­th false negative SARS-CoV-2 testing at hospital admission: A retrospective multi-center study

PONE-D-20-38011R1

Dear Dr. Simonov,

We’re pleased to inform you that your manuscript has been judged scientifically suitable for publication and will be formally accepted for publication once it meets all outstanding technical requirements.

Kind regards,

Chiara Lazzeri

Academic Editor

PLOS ONE
---

## [Editor Report · Acceptance letter]

3 May 2021

PONE-D-20-38011R1 

Predicting patients wi­­­th false negative SARS-CoV-2 testing at hospital admission: A retrospective multi-center study 

Dear Dr. Wilson:

I'm pleased to inform you that your manuscript has been deemed suitable for publication in PLOS ONE. Congratulations! Your manuscript is now with our production department. 

Kind regards, 

on behalf of

Dr. Chiara Lazzeri 

Academic Editor

PLOS ONE